# Linking Affect Dynamics and Well-Being: A Novel Methodological Approach for Mental Health

**DOI:** 10.3390/healthcare12171690

**Published:** 2024-08-24

**Authors:** Gloria Simoncini, Francesca Borghesi, Pietro Cipresso

**Affiliations:** Department of Psychology, University of Turin, 10124 Turin, Italy; gloria.simoncini@unito.it (G.S.); pietro.cipresso@unito.it (P.C.)

**Keywords:** affective states, affect dynamics, psychometrics, well-being, electromyography, depression, cognitive distortions

## Abstract

Emotions are dynamic processes; their variability relates to psychological well-being and psychopathology. Affective alterations have been linked to mental diseases like depression, although little is known about how similar patterns occur in healthy individuals. This study investigates the psychophysiological correlations of emotional processing in healthy subjects, specifically exploring the relationship between depressive traits, cognitive distortions, and facial electromyographic (f-EMG) responses during affective transitions. A cohort of 44 healthy participants underwent f-EMG recording while viewing emotional images from the International Affective Picture System (IAPS). Self-report measures included the Beck Depression Inventory (BDI) and the Cognitive Distortion Scale (CDS). Higher BDI scores were associated with increased EMG activity in the corrugator muscle during transitions between positive and negative emotional states. Cognitive distortions such as Catastrophizing, All-or-Nothing Thinking, and Minimization showed significant positive correlations with EMG activity, indicating that individuals with higher levels of these distortions experienced greater facial muscle activation during emotional transitions. This study’s results indicate that there is a bidirectional correlation between depressed features and cognitive distortions and alterations in facial emotional processing, even in healthy subjects. Facial EMG in the context of dynamic affective transitions has the potential to be used as a non-invasive method for detecting abnormal emotional reactions at an early stage. This might help in identifying individuals who are at risk of developing depression and guide therapies to prevent its advancement.

## 1. Introduction

The average emotional state of an individual may provide valuable insights into their overall state of well-being and psychological adaptation [1]; alterations represent core signs of several mental disorders [2]. Nevertheless, emotions are not static conditions, but rather dynamic processes that vary in response to circumstances and regulatory influences [1,3]. Contemporary procedures emphasize the dynamic and evolving nature of emotions, proposing that emotional experiences arise from the interplay of many elements and undergo constant transformation [4]. Recently, the affect dynamics field tried to map emotional transitions and dynamics [4] via the application of the Experience Sampling Method (ESM)—a methodology that allows the collection of intensive longitudinal data of people’s subjective experiences by using brief mobile questionnaires [5,6]. The relationship between affect dynamics, well-being, and mental health is complex and multifaceted: a higher and sudden emotional variability and instability—rapid emotional changes—is linked to lower psychological well-being and greater psychopathology; similarly, a higher level of inertia, which refers to the persistence of emotional states—especially negative ones—over time, can be associated with a decline in mental health [7,8]. Indeed, individuals with elevated degrees of neuroticism, depression, and other forms of psychopathology experience frequent and unpredictable changes in their emotional states [9,10]. Recent studies investigated how rumination, life stress [11], and daily affective dynamics [12] predict the course of depression over time. Their findings highlighted that individuals with depression exhibit distinct affective dynamics compared to healthy controls; specifically, depressed individuals showed elevated daily negative affect (NA) and NA variability, along with decreased positive affect (PA) [12]. Moreover, Momentary Ruminative Self-Focus (MRS) in response to stress predicted greater prospective increases in depressive symptoms in healthy subjects [11].

Traditionally, depression is marked by unreasonable and biased thinking, referred to as cognitive distortions, which might exacerbate and perpetuate depressed symptoms. They include unfavorable cognitive habits that are usually involuntary and could affect an individual’s perception of reality [13]. These distortions can drive people to develop catastrophic thinking, letting them imagine the worst possible result of a scenario, regardless of its likelihood. These processes can additionally lead people to minimize positive occurrences and to overstate self-responsibility [14]. The massive adoption of cognitive distortions does not necessarily imply the presence of a diagnosed depressive disorder; in fact, these are borderline conditions, in which the symptoms are defined as “subclinical”. Subclinical depression, which does not match the diagnostic criteria for Major Depressive Disorder (MDD), may still have a significant effect on emotional processing and expression. Patients with subclinical depression showed milder but still noticeable changes in EMG activity, indicating possible early signs of emotional dysregulation [15]. Benning and Ait Oumeziane [16] showed that individuals with subclinical depression had fewer positive emotional responses, as shown by reduced activity in the zygomaticus major muscle when exposed to pleasant stimuli. This result aligns with the low positive emotion hypothesis of depression, which suggests that a reduced ability to experience pleasant feelings is an essential aspect of depressed symptoms. Comparative examinations of patients with subclinical and severe depressive states indicate that, whereas both groups showed altered facial EMG patterns, the degree and features of these changes varied [17,18]. Individuals with subclinical depression had milder but still noticeable changes in EMG activity, indicating possible early signs of emotional dysregulation. Patients suffering from MDD, on the other hand, demonstrated more substantial declines in their capacity to convey emotions via facial expression [17].

Although psychophysiological components appear to be important in the description of symptoms, particularly subclinical ones, there is no research that examines their link in terms of affect dynamics. Indeed, a combination of self-reported and physiological data, within a framework that takes into account emotional variability, could offer a more comprehensive view of how individuals experience and react to emotional situations in real time [19,20]. For this reason, by combining information from research on affect dynamics and deriving from studies of facial electromyography and depressive symptoms, the present study seeks to create a bridge between these areas through a new methodological formulation. Furthermore, given the previous observation that alterations in facial electromyography (fEMG) activity occur not only in diagnosed cases of depression but also in more mild circumstances [21,22,23], we are interested in determining whether it is feasible, through this methodology, to identify any alterations at an even earlier stage.

Among the psychophysiological measures, the use of fEMG in the research of depressive symptoms is now widely recognized. fEMG is a non-invasive method that allows for a detailed exploration of the complex connection between face muscle activity and emotional regulation and expression [24]. fEMG is a methodology employed to detect and record the electrical signals produced by facial muscles since it is capable of tracking even small and subtle changes occurring in response to different emotional stimuli. EMG’s sensitivity to micro-expressions makes it a great resource for evaluating emotional reactions that are not readily evident [24,25,26,27,28] and it can be used to differentiate various strategies for regulating emotions. Recent research has shown that individuals with major depressive disorder (MDD) have reduced activity in the zygomaticus major muscle, which is associated with pleasant emotions like smiling [29]. Kim and colleagues [29] used the Facial Action Coding System (FACS) to identify distinct facial expression patterns in older adults with higher levels of depressive symptoms. These patterns include downward and inward pulls at the corners of the mouth during posed expressions and raised and narrowed inner brows during spontaneous expressions. These findings suggest that facial expressions could be useful markers for detecting depression symptoms in older individuals. Hill et al.’s [30] research also corroborated the emotion–context insensitivity paradigm, suggesting that sadness causes a broad decrease in emotional reactivity in both positive and negative settings.

Studying subclinical depressive symptoms offers considerable insight into how these frequently overlooked forms of mood disorder affect the way emotions are processed. It is crucial to have a thorough grasp of the nuanced variations in fEMG patterns in individuals exhibiting depressive-like symptoms while reacting to emotional stimuli for several significant reasons. It improves the timely identification and management of mild symptoms, perhaps preventing their escalation into severe depressive episodes. Moreover, it offers a deeper comprehension of the emotional dysregulation that characterizes depression, which may contribute to the development of targeted intervention approaches [31].

The methodological framework adopted found its theoretical basis in Russell’s Circumplex Model of Affect [32]. Russell’s Circumplex Model organizes emotions based on two dimensions: arousal and valence. The arousal dimension represents the level of intensity of an affective state, and it ranges from low to high arousal; the valence dimension represents the degree of positivity or negativity of an affective state, ranging from unpleasant to pleasant [32]. By combining these two dimensions, four affective conditions arise. These states are characterized by intense arousal and negative valence (for example, stress), intense arousal and positive valence (e.g., excitement), reduced arousal and negative valence (e.g., boredom), and reduced arousal and positive valence (e.g., relaxation) [33]. In order to enhance the granularity of our analysis, we have directed our attention to the transitions between different affective states, rather than focusing on stable emotional states, since the individual’s emotional experience in everyday life is indeed characterized by affect dynamics [4]. To maintain an ecological viewpoint, it was decided to reproduce, inside a controlled laboratory environment, all the possible emotional variations that individuals might experience in their daily lives. Thus, referring to Russell’s model, the four affective states provided a basis for identifying 12 potential transitions that may occur between them. Furthermore, alongside the transition indexes, state–trait indexes were also selected for extrapolation. These measures represent the subject’s psychophysiological activity when experiencing a particular affective state. By observing these indexes, we may get information on the consistency of the emotional states that the individual is experiencing.

To evaluate the feasibility of this methodology that we proposed, the decision was made to carry out the study on a group of healthy subjects, with a specific emphasis on mental health prevention. Therefore, we choose to use an exploratory approach to discover whether it is possible to detect comparable patterns found in prior research on depression [21,30,34] in individuals without depression but who exhibit varying levels of depressive-like features. Moreover, in certain circumstances, the lack of objective data, such as psychophysiological measures, may restrict the ability to effectively understand the relationship between affect dynamics, physiological correlates, and potential symptomatology. Indeed, a combination of self-reported and physiological data, within a framework that takes into account emotional variability, could offer a more comprehensive view of how individuals experience and react to emotional situations in real time.

## 2. Materials and Methods

### 2.1. Participants

The invitation to participate in this study was conducted using both probabilistic cluster sampling and avalanche sampling methods. The trial organizers reached out to the volunteer participants who had consented to participate in the study via email to coordinate the specific time and place of the experiment. The experimental activity took place in the HST laboratory of the Psychology Department (University of Turin). A group of 44 individuals, consisting of 25 women, willingly participated in the experiment. The average age of the males was 26 with a standard deviation of 3.59 (ranging from 21 to 36), while the average age of the women was 24.77 with a standard deviation of 4.99 (ranging from 18 to 43). Both gender groups have a unimodal age distribution, with a single noticeable peak occurring in the mid-20s. The minimum sample size required to test the study hypothesis was determined using G*Power version 3.1.9.7 [29] through a priori Power Analysis, based on the effect size of a previous study [21]. Specifically, Tan et al. (2012) found that monitoring the activity of the corrugator supercilii and zygomaticus major muscles in facial EMG in a sample of healthy subjects could reliably discriminate between various emotional states. Results indicated the required sample size to achieve 95% power for detecting a medium effect size of 0.52 [21], at a significance criterion of α = 0.05, was 42 participants.

The research was conducted in accordance with the Declaration of Helsinki, after permission from the Ethics Committee of Istituto Auxologico Italiano Piancavallo (Prot. No. 2022_10_25_05).

### 2.2. Inclusion Criteria

Before taking part in the experiment, all individuals gave written informed consent. This study’s inclusion criteria were that participants must be a minimum of 18 years old and have no reported mental illnesses. Additionally, it was essential that participants did not have any cardiovascular, neuromotor, neurological problems, or cardiac arrhythmias.

### 2.3. Procedure

The experimental procedure was partitioned into two distinct phases (see Figure 1).

#### 2.3.1. Self-Report Measures

In the first phase, participants completed two self-report questionnaires, lasting approximately 5 min. They consisted of two scales aimed at assessing the presence of depressive symptoms and cognitive distortions:The Beck Depression Inventory (BDI) [35] is a widely used instrument for assessing the severity of depressive symptoms and it has been adapted for use as a screening tool in the general population. It consists of 21 items (scored on a scale from 0 to 3); each item on the BDI consists of four statements, ranked in order of increasing severity, describing common symptoms of depression. The minimum obtainable score equals 0, while the maximum corresponds to 63. Respondents choose the statement that best describes how they have been feeling over the past week.The Cognitive Distortion Scale (CDS) [36] is designed to assess cognitive distortions, which are biased ways of thinking that are believed to contribute to psychological distress. This scale measures the frequency of cognitive distortions across two domains: interpersonal and achievement-related situations. The CDS consists of 20 items, divided into 10 categories and rated on a 7-point Likert scale (1 = Never, 7 = Always). Each category consists of 2 items, resulting in a minimum observable score of 2 and a maximum score of 14 for each category. Regarding the total scale (CDS), scores between 20 and 140 can be observed.

#### 2.3.2. Psychophysiological Evaluation

During the second phase, participants were introduced to the experiment and instructed to sit in the observation laboratory of the Department of Psychology. During the experiment, participants were directed to focus on emotional images that were displayed randomly on a computer monitor. The images were chosen from the International Affective Picture System (IAPS) database, which is an organized set of pictures developed with the intention of evoking emotional responses [37]. The IAPS pictures are categorized as pleasant, neutral, or unpleasant, and employed to induce affective states in a static manner. The Arousal and Valence of each IAPS picture were determined using the 9-point Likert scale of the Self-Assessment Manikin (SAM), as established by Bradley and Lang in 1994 [38]. The selection of the images has been driven by Russell’s Circumplex affect model [32] (Figure 1). The model comprises four quadrants, which are determined by the point where arousal and valence levels overlap. Quadrant “A” generally represents the state of “Stress”, characterized by high arousal and negative valence. Quadrant “B” represents the state of “Engagement”, defined by high arousal and positive valence. Quadrant “C” represents the state of “Boring”, marked by low arousal and negative valence. Quadrant “D” represents the state of “Relax”, defined by low arousal and positive valence. The transitions among the four quadrants entail shifts in the horizontal, vertical, and diagonal axes. Horizontal transitions occur along the valence axis, indicating changes between positive and negative emotional states. Conversely, vertical transitions involve shifts in levels of arousal, ranging from low activation to high activation. Lastly, oblique transitions impact both arousal and valence, (e.g., moving from low arousal and negative valence to high arousal and positive valence, or vice versa, and from low arousal and positive valence to high arousal and negative valence, or vice versa).

Consequently, we selected from the IAPS images that possessed a significant degree of arousal/valence for Condition B (High arousal and Positive Valence), as established by Likert point ratings above 6, and a minimal degree of arousal/valence, as defined by Likert point ratings below 4, for Condition C (Negative valence or Low arousal). Images were thereafter included randomly in 13 blocks. A block was composed of 12 images with a 10-second latency each; in this way, every transition lasted 240 s (120 s for each block) [39,40]. To prevent a temporal or ceiling impact on their emotional reaction, each participant saw a different series of randomly selected pictures. This method avoided order effects by presenting the images in a randomized order to allow for a thorough investigation of emotional transitions.

Since our intention was to focus on emotional transitions, we considered only that the 30” transition between two blocks cuts were made on the various blocks so that the 30 s between transitions (last 15 s of the previous block and first 15 s of the next block) were considered (Figure 2). In addition to this, in order to explore the internal average activation of each quadrant, corresponding to the state–trait indexes (named, AA, BB, CC, DD), the net of the transition phase and the middle 30 s of each block were extrapolated (thus included between second 45 and second 75) [39,40].

#### 2.3.3. EMG Recording and Signal Processing

This study used 4 mm diameter bipolar skin surface electrodes to capture facial EMG activity. The electrodes were positioned on the corrugator supercilii (EMG1) and zygomaticus major muscle (EMG2) regions on the left side of the participants’ faces, following the EMG placement parameters suggested by Fridlund and colleagues [41].

Then, a brief 2 min initial session was carried out with all participants in order to establish a consistent and reliable reference value. The psychophysiological evaluation began upon a specific trigger that was coordinated with the experimental stimuli. After a debriefing session, the researcher assisted the subjects in removing all electrodes and patches. The EMG signals were acquired at a sampling rate of 1024 Hz to provide enough range of variability for the signal processing of this intricate muscle activity. Signals were recorded using Nexus-10 and then processed using custom software in MATLAB 9.13.0 (R2022b) [42]. A 50 Hz notch filter was applied to remove power line interference. Subsequently, amplitude filtering was performed within the 20–500 Hz range. Additionally, a visual inspection was conducted to identify and remove any remaining artifacts.

Two individuals reported a highly filthy signal with electrical noise because the psychophysiological fEMG signal, which emerged during the processing phase, was incorrectly recorded. As a result, they were regarded as missing data and included in Table 2 as “missing”.

The main indicator adopted for the investigation was the amplitude of the EMG signal, which reflects the extent of muscle activity, while participants were exposed to emotional stimuli. Greater amplitude values imply higher levels of muscular activation, indicating an enhanced emotional reaction to stimuli [43]. Through the investigation of EMG signals’ amplitude in response to emotional transitions, the objective was to determine the relationship between positive or negative affective responses and the psychological traits assessed with the self-report questionnaires. Specifically, we used the ratio EMG1/EMG2 since it allowed us to quantify the relative activity levels of the two different muscle groups, thus providing insights into their balance. In fact, a lower ratio indicates a predominance of positive emotional states, where zygomaticus activity exceeds corrugator activity. Conversely, a higher ratio suggests a dominance of negative emotional states, where corrugator activity surpasses zygomaticus activity. When the activities are approximately equal, the ratio approaches one, indicating a neutral or mixed emotional state. 

#### 2.3.4. Statistical Analyses

The analyses were conducted using Jamovi Statistics software, version 2.3.28. To investigate the connection between physiological emotional activation (EMG1/EMG2) during transitions and specific mood traits, we calculated a correlation matrix using Spearman’s *ρ* correlation coefficient given the non-normality of data distribution.

Confidence intervals were included in the correlation matrices to provide an additional indication of the significance of the results.

## 3. Results

We employed a correlation matrix to explore the association between mood features and emotional responses, as indicated by the ratio of facial EMG amplitudes of EMG1 and EMG2. Descriptive tables of self-report questionnaires and EMG measures are reported below (Table 1 and Table 2).

To analyze the directionality of electromyographic activation during different transitions, Figure 3 displays the EMG activity inside each block. It allowed us to observe and compare the increase or decrease in activity across consecutive blocks. The color of the boxes in the figure indicates that the increase and decrease in activation within the 13 blocks, when evaluated in pairs, are in accordance with the associated quadrant (A, B, C, or D). The EMG1/EMG2 ratio’s values rise when the valence is negative and reduce when it is positive.

In our study, we analyzed the data using a correlation matrix. This statistical approach was chosen due to its suitability for our experimental designs and for the explorative nature of our research. Only significant correlations between mood traits and physiological components were listed in Table 3 and Table 4 (non-significant correlations can be observed in Appendix A).

The analysis of the correlation matrix revealed significant relationships between various self-report measures and facial EMG transitions.

The correlation between the BDI and AB was *ρ* = 0.349 (*p* < 0.05) and with CD was *ρ* = 0.314 (*p* < 0.05). Similarly, the BDI sum correlated positively with DB (*ρ* = 0.364, *p* < 0.05), DA (*ρ* = 0.320, *p* < 0.05), BC (*ρ* = 0.319, *p* < 0.05), and CB (*ρ* = 0.383, *p* < 0.05) transitions, reflecting consistent moderate associations across these variables.

Catastrophizing, a sub-component of the CDS, shows notable correlations with multiple EMG ratio transitions. The correlation with AB (from “stress” to “engagement”) was *ρ* = 0.349 (*p* < 0.05), demonstrating a moderate positive relationship. Similar correlations were observed with BD (*ρ* = 0.383, *p* < 0.05), DA (*ρ* = 0.333, *p* < 0.05), and CB (*ρ* = 0.368, *p* < 0.05).

The measure of All-or-Nothing Thinking also presented a significant correlation with BD transition (*ρ* = 0.347, *p* < 0.05).

Should Statements scores correlated moderately with several EMG ratio transitions. Specifically, the correlation with CD transition was *ρ* = 0.316 (*p* < 0.05), with DC transition was *ρ* = 0.332 (*p* < 0.05), with CA was *ρ* = 0.307 (*p* < 0.05), with DB *ρ* = 0.356 (*p* < 0.05), and with CB was *ρ* = 0.358 (*p* < 0.05).

The CDS measure of Minimization displayed significant correlations with several transitions. The correlation with CD was *ρ* = 0.364 (*p* < 0.05), with DC was *ρ* = 0.322 (*p* < 0.05), with DA was *ρ* = 0.386 (*p* < 0.05), with BC was *ρ* = 0.312 (*p* < 0.05), and with CB was *ρ* = 0.304 (*p* < 0.05). The strongest correlation was observed with DB transition (*ρ* = 0.407, *p* < 0.01).

CDS total score showed significant correlations with three affective transitions: with AB (*ρ* = 0.373, *p* < 0.05), with DB (*ρ* = 0.444, *p* < 0.01), and with CB (*ρ* = 0.318, *p* < 0.05).

In addition to the previously analyzed correlations, we have further significant relationships between various self-report measures and EMG state–trait transitions. These results extend our understanding of the interconnectedness between psychological constructs and physiological states.

BDI scores displayed a correlation with DD state–trait (*ρ* = 0.312, *p* < 0.05).

The correlation between the CDS sum and DD state–trait was *ρ* = 0.373 (*p* < 0.05).

Should Statements (CDS) show a more consistent correlation with DD state–trait, with *ρ* = 0.426 and *p* < 0.01.

Minimization (CDS) also shows significant correlations with BB and DD transitions. The correlation with BB was *ρ* = 0.383 (*p* < 0.05), and with DD was *ρ* = 0.391 (*p* < 0.05).

Catastrophizing (CDS) scores emerged correlated with two state–trait conditions: the first one is BB (*ρ* = 0.372, *p* < 0.05) and the second one is DD (*ρ* = 0.352, *p* < 0.05); both correlations reflected a moderate positive relationship.

All-or-Nothing Thinking demonstrates significant correlations with DD state–trait (*ρ* = 0.444, *p* < 0.01), indicating a stronger positive association.

## 4. Discussion

This study examined possible interactions between certain mood features and physiological emotional response, using facial electromyography. The idea was to study specific mood dispositions often associated with physiological correlates at an early stage and with healthy subjects.

The findings highlight the intricate connections between psychological constructs and physiological responses, offering valuable insights for further exploration. First of all, it was found that higher scores on the BDI scale correlated positively with the EMG1/EMG2 ratio of five emotional transitions, specifically with CD (transition from poorly activating negative emotional states to poorly activating positive emotional states, e.g., from boredom to relaxation), DB (transition from poorly activating positive emotional state to highly activating positive emotional state, e.g., from relaxation to arousal), DA (transition from poorly activating positive emotional state to highly activating negative emotional state, e.g., from relaxation to stress), BC (from highly activating positive emotional state to poorly activating negative emotional state, e.g., from arousal to boredom), and CB (transition from poorly activating negative emotional state to highly activating positive emotional state, e.g., from boredom to arousal). What stands out here is the greater activation of the corrugator muscle, typically associated with negative emotions, compared to the zygomatic in situations of transition between strongly activating positive states and slightly activating negative states. Specifically, it seemed like negative states predominated over positive ones during transitions, thus contributing to maintaining a higher activation of the corrugator muscle. The findings are consistent with the literature on depressive disorders and patterns of psychophysiological activation [16,17,34] but also with findings from Ecological Momentary Assessment (EMA) research, which revealed a predominance of negative affect as self-perceived by the subjects [11,44,45].

Furthermore, considering that depressive symptoms tend to be associated with certain cognitive distortions [46], it appeared suitable to investigate the link between these distortions and the EMG1/EMG2 ratio during emotional transitions. Notable correlations have been found between the overall score on the CDS scale and psychophysiological activations during three transitions: AB (transition from stress to engagement), DB (transition from relaxation to engagement), and CB (transition from boredom to engagement). Once more, the presence of extremely activating emotional states is noticeable (B). Prior studies in the literature [47,48] have shown the association between individuals with depression who engage in cognitive distortions and the manifestation of alterations in facial muscle activation. These modifications are often associated with the automatic processing of emotional inputs, whereby individuals with depression tend to process negative facial expressions more quickly and readily, whereas delighted expressions take more cognitive effort and are less likely to be performed [48]. The CDS scale investigates ten distinct cognitive distortion processes that can occur independently of each other and whose sum generates the final score. Given that the sample included healthy individuals, it is crucial to thoroughly analyze the potential connections that may arise between mood characteristics and psychophysiological activity. This is because only some, but not all, of the cognitive distortion processes are expected to be present in healthy subjects [49]. Therefore, we also decided to evaluate the scores achieved for the ten distinct cognitive distortion processes independently.

Catastrophizing refers to a cognitive distortion involving making pessimistic predictions about the future without any empirical foundation. This cognitive pattern prompts people to anticipate the most unfavorable consequences, often intensifying their adverse emotional reactions and worry [47]. The scores on the “Catastrophizing” subcomponent of the CDS scale positively correlate significantly with the EMG1/EMG2 ratio in transitions: AB (transition from a very activating negative emotional state to a very activating positive emotional state, e.g., from stress to excitement), BD (from a highly activating positive emotional state to a slightly activating positive emotional state, e.g., from engagement to relax), DA (from a slightly activating positive emotional state to a very activating negative emotional state, e.g., from relaxation to stress) and CB (from a slightly activating negative emotional state to a very activating positive emotional state, e.g., from boredom to excitement). What emerged is that greater levels of Catastrophizing were associated with greater activation of the corrugator muscle compared to the zygomaticus during the transition between emotionally stressful or negative stimuli and very activating positive stimuli (e.g., AB, DA, CB) but also between positive states (BD). There seemed to be a tendency to prioritize negative stimuli rather than positive ones during transitions by maintaining an attitude of inertia toward negative effects; this attitude could highlight the nature of this distortion mechanism through the orientation towards negative stimuli [47,50] even in relation to positive stimuli.

Dichotomous thinking, also referred to as All-or-Nothing Thinking, is a cognitive distortion characterized by the inclination to interpret events or circumstances in extreme, polarized terms. This entails experiencing a reference class only in connection to the extreme ends of each spectrum, so ignoring any neutral or moderate aspects of occurrences and interpreting them as either completely positive or entirely adverse [51]. In the present study, this cognitive distortion had a positive correlation with the EMG1/EMG2 ratio during one transition, BD, that includes the passage from a highly activating positive emotional state to a slightly activating positive emotional state (e.g., from excitement to relaxation). Due to the tendency of All-or-Nothing Thinking, individuals may interpret situations in a binary manner, without considering the subtle differences in between. This trait can make it challenging to understand the transitions between nuanced positive emotional states, such as in the case of BD transitions.

The term “Should Statements” refers to a cognitive distortion where individuals impose strict rules or expectations on themselves or others, often leading to feelings of frustration, guilt, or inadequacy when these expectations are not met [52]. “Should Statements” scores turned out to be positively correlated with EMG1/EMG2 ratio during five transitions: CD (from a slightly activating negative emotional state to a slightly activating positive emotional state, e.g., from boredom to relax), DC (from a slightly activating positive emotional state to a slightly activating negative emotional state, e.g., from relax to boredom), CA (from a slightly activating negative emotional state to a highly activating negative emotional state, e.g., from boredom to stress), DB (from a slightly activating positive emotional state to a highly activating positive emotional state, e.g., from relax to excitement) and CB (from a slightly activating negative emotional state to a highly activating positive emotional state, e.g., from boredom to engagement). In this context, the presence of “Should Statements” as a trait seems to be connected to negative emotional states (as shown by greater activation of the corrugator muscle) during transitions affecting a single axis: either that of arousal (e.g., CA, DB) or that of valence (e.g., CD, DC). These emotional transitions, which occur along a single axis, are inherently more subtle and gradual, resulting in fewer abrupt changes and a reduced level of reaction for the individual involved. The observed pattern could be attributed to the influence of Should Statements on attention toward external stimuli. This influence is characterized by an increase in cognitive load, reinforcement of negative emotional states, and promotion of a negative cognitive bias. These factors collectively hinder an individual’s ability to effectively engage with and respond to their external environment [52,53].

Minimization is a cognitive bias in which a person underestimates or diminishes the positive value of an event, emotion, or circumstance. This psychological phenomenon often manifests as a protective strategy to avoid facing painful feelings or realities [47]. Higher Minimization scores emerged to be positively correlated with higher EMG1/EMG2 ratio during six transitions: CD (transition from low activating negative emotional state to low activating positive emotional state, e.g., from boredom to relaxation), DB (from low activating positive emotional state to highly activating positive emotional state, e.g., from relax to excitement), DC (from low activating positive emotional state to low activating negative emotional state, e.g., from relax to boredom), DA (from low activating positive emotional state to highly activating negative emotional state, e.g., from relax to stress), BC from highly activating positive emotional state to poorly activating negative emotional state, e.g., from arousal to boredom), and CB (transition from low activating negative emotional state to highly activating positive emotional state, e.g., from boredom to excitement). What emerged as particularly interesting is the maintenance of greater activation of the corrugator muscle, linked to negative emotions during transitions from negative to positive states and vice versa. It is possible that a general tendency to minimize the positive aspects of stimuli and emotional states played a role in all this, contributing to an anchorage to negative emotional responses even in the occurrence of extremely positive experiences [54].

Ultimately, a distinct pattern became evident while examining the correlation matrices that represented the state–traits. The overall score on the BDI scale was shown to be correlated with the EMG1/EMG2 ratio in the BB condition, which corresponds to a positive emotional state with a high level of activation. The total score of the CDS scale exhibited a substantial association with the EMG1/EMG2 ratio in the DD condition, which indicated an amount of variability among relaxation circumstances. A similar pattern has been identified for the component of “Minimization” in both the BB and DD quadrants. In our study, “All-or-Nothing Thinking” had a positive correlation with EMG1/EMG2 ratio during BB states. What emerged is consistent with the previous literature emphasizing the characteristics of rigidity, disappointment, and loss perceived by individuals who implement this cognitive distortion [55].

These findings once again emphasized the link between certain trait features and emotional responses to specific stimuli, as well as the maintenance of a pattern of inertia that generated a prevalence of electromyographic activation associated with negative emotions (fEMG1) even and especially under positive affective conditions, suggesting an inaccurate representation of the emotional valence of these stimuli [56].

Our results emphasized the connection between certain mood traits and specific patterns of fEMG activation in a cohort of healthy subjects. The prevalent activation of the corrugator muscle even in the presence of positive emotional stimuli has highlighted alterations that could be associated with underlying cognitive-emotional processes, such as emotion regulation (ER) [53]. What emerged as significant was not only the predominance of negative emotional expressions over positive ones during transitions, but also how this attitude was correlated with depressive-like mood patterns and cognitive distortions especially along with positive emotional states.

By employing the proposed methodology, we were able to investigate emotional states by capturing their dynamic nature; furthermore, through the use of fEMG, we managed to detect distinct patterns of facial muscle activity in response to changes in affective states, specifically in terms of arousal and valence. Ultimately, this study examined the associations between several mood characteristics and trends in fEMG activity during affective transitions. Previous studies have established a strong connection between heightened activation of the corrugator muscle in response to positive and negative emotional states and the occurrence of mild and severe depressive disorders [16,17,30]. However, there has been a smaller emphasis on examining depressive features among healthy subjects; actually, in order to prioritize mental health prevention, it is essential to thoroughly examine these conditions, particularly. Early alterations in facial expressiveness have indeed been detected in individuals at risk for psychosis [23,57]. Examining emotional expressiveness in response to affective transitions might be a valuable approach to identifying individuals who may be at risk of developing depressive symptoms, allowing for prevention and appropriate intervention [58,59,60].

## 5. Conclusions

The present study is within the continuum of affect dynamics, well-being, and the impact that psychopathological, and more precisely depressive, symptoms have on individual emotional processing. Research in this field has mainly focused on mild and severe depressive conditions, without however focusing on previous stages, observing such attitudes in healthy subjects. Previous studies have revealed the connection between depressive symptoms and facial electromyographic alterations in correspondence with emotional stimuli, outlining difficulties in the emotional regulation processes of these individuals [1,29,30,34].

We proposed a method that combined self-report questionnaires with psychophysiological measures, such as fEMG. While self-report methods may be prone to reporting bias and dependent on the subject’s awareness and willingness to express their emotional states, the inclusion of fEMG within the diagnostic process could provide a more objective assessment [61]. This approach could be particularly valuable in identifying minor symptoms and sub-clinical conditions that might not be fully captured through questionnaires alone. The employment of fEMG, which has been linked to both clinical and sub-clinical depression, could enhance the detection of early signs or symptoms that might otherwise go unnoticed.

The findings of our study could have implications for the early detection and prevention of mental health issues. The observed associations between specific cognitive distortions and altered EMG activity during emotional transitions suggest that facial EMG could serve as a non-invasive instrument for identifying some risk patterns in terms of psychopathology. In fact, a consistent activation of the corrugator muscle emerged, associated with negative emotions, in multiple affective transitions also involving positive stimuli. Furthermore, these alterations were found to be related to specific cognitive distortions (e.g., Catastrophizing, Minimization, and Should Statements) and depressive traits. The early identification of atypical psychophysiological responses towards dynamic emotional stimuli, together with higher levels of cognitive distortions, could facilitate timely interventions [62], potentially preventing the progression to more severe depressive states. Additionally, the insight gained from this study could inform the development of targeted therapeutic approaches aimed at modifying maladaptive cognitive patterns and enhancing emotional regulation skills.

## 6. Limitations

This study presents some limitations, primarily due to static emotional stimuli (IAPS Images), that do not properly represent the dynamic nature of emotional experiences. More fluid and continuous stimuli might enhance transitions between different emotional states [63]. It is also necessary to mention that the effect sizes observed in the present study are smaller (in most cases, *ρ* ≈ 0.3) than those used to establish the sample size in the Power Analysis (r = 0.52) and CI due to intraindividual variability in psychophysiological parameters. Future studies should consider these limitations and address them to enhance statistical power.

The current study’s exploratory characteristics provide a novel viewpoint on the examination of potential early signs of psychopathology; however, it needs further validation via longitudinal investigations to confirm the findings. Moreover, in order to maintain an ecological research framework, a future application could employ ESM paradigms [64]. While they may not capture all the emotional states and transitions the subject experiences, using non-invasive psychophysiological measures, such as heart rate variability, may provide objective data to corroborate the participant’s subjective responses [65].

## Figures and Tables

**Figure 1 healthcare-12-01690-f001:**
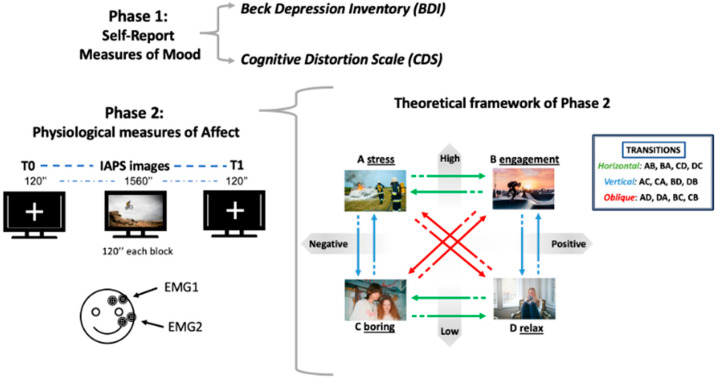
Graphical depiction of Phase 1: self-report measures of mood; and Phase 2: psychophysiological measures of affect using facial EMG (positioned on the corrugator supercilii and the zygomaticus muscle). On the right, the theoretical framework of Phase 2 is represented.

**Figure 2 healthcare-12-01690-f002:**
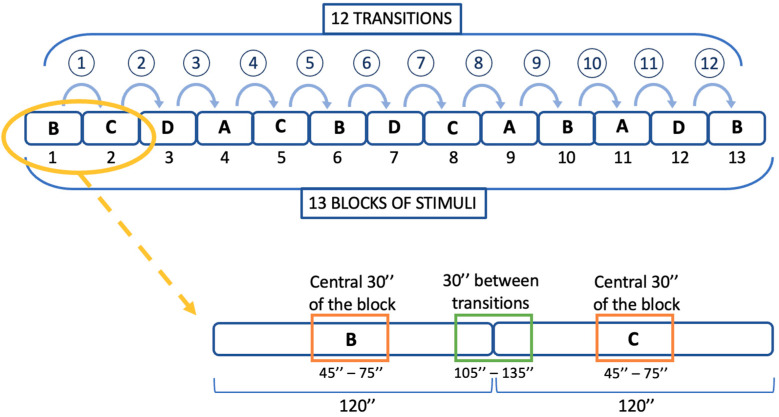
Graphical representation of the 12 transitions across the 13 blocks, with a close-up of the cutting process. Stress is represented by A blocks, Excitement is represented by B blocks, Boredom is represented by C blocks, and Relaxation is represented by D blocks.

**Figure 3 healthcare-12-01690-f003:**
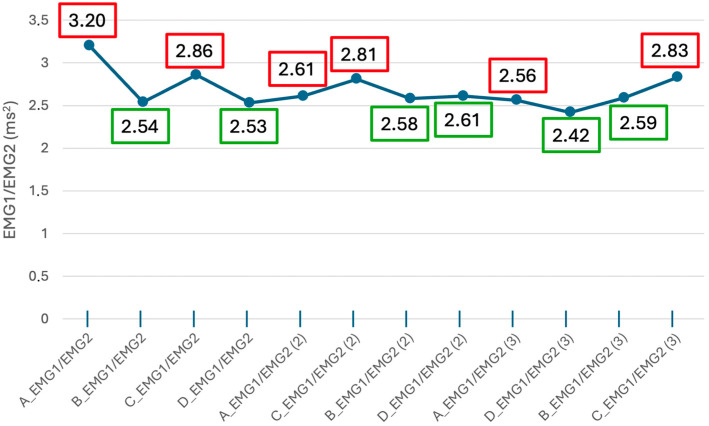
Amplitude EMG1/EMG2 ratio within each block (green boxes represent positive valence states; red boxes represent negative valence states), based on Russell’s Circumplex model. A blocks represent Stress, B blocks represent Excitement, C blocks represent Boredom, and D blocks represent Relaxation.

**Table 1 healthcare-12-01690-t001:** Descriptive of self-report questionnaires.

Construct	Type of Measurement	Questionnaires ^1^	N	Mean	SD	Min	Max	Cronbach’s Alpha	Skewness		Kurtosis	
									Skewness	SE	Kurtosis	SE
Depression and Cognitive Distortion	Self-Report	Mind Reading (CDS)	44	9.45	2.80	3	14	0.81	−0.42	0.36	−0.58	0.70
		Catastrophizing (CDS)	44	7.50	3.36	2	14	0.82	0.28	0.36	−0.93	0.70
		All or nothing thinking (CDS)	44	6.68	2.73	2	13	0.53	−0.04	0.36	−0.45	0.70
		Emotional reasoning (CDS)	44	8.18	3.12	2	14	0.90	−0.16	0.36	−1.11	0.70
		Labeling (CDS)	44	6.89	2.86	2	13	0.79	0.24	0.36	−0.57	0.70
		Mental filter (CDS)	44	7.30	3.08	2	14	0.86	0.32	0.36	−0.81	0.70
		Hypergeneralization (CDS)	44	6.75	3.27	2	14	0.74	0.19	0.36	−1.12	0.70
		Personalization (CDS)	44	7.27	2.57	2	13	0.72	0.49	0.36	−0.25	0.70
		Doverization (CDS)	44	8.95	2.79	2	14	0.58	−0.51	0.36	−0.44	0.70
		Minimization (CDS)	44	7.30	2.82	2	13	0.78	0.15	0.36	−0.76	0.70
		CDS	44	76.27	20.84	29	117	0.92	−0.24	0.36	−0.30	0.70
		BDI	44	9.02	7.74	0	31	0.89	1.38	0.36	1.69	0.70

^1^ Acronyms: BDI stands for Beck Depression Inventory; CDS refers to Cognitive Distortion Scale.

**Table 2 healthcare-12-01690-t002:** Descriptive of physiological measures during transitions and state–trait.

Type of Transition	Physiological Measures ^1^	N	Missing	Mean	SE	95% Confidence Interval		Min	Max	Skewness		Kurtosis	
						Lower	Upper			Skewness	SE	Kurtosis	SE
Horizontal	AB_EMG1/EMG2	42	2	2.89	0.22	2.44	3.34	0.97	6.92	1.17	0.36	1.24	0.72
	BA_EMG1/EMG2	42	2	2.62	0.25	2.13	3.12	0.56	8.93	1.53	0.36	4.59	0.72
	CD_EMG1/EMG2	42	2	2.73	0.24	2.25	3.21	0.21	8.09	1.15	0.36	2.46	0.72
	DC_EMG1/EMG2	42	2	2.57	0.23	2.10	3.05	0.47	8.42	1.54	0.36	3.83	0.72
Vertical	AC_EMG1/EMG2	42	2	3.07	0.24	2.58	3.56	0.79	8.78	1.10	0.36	2.56	0.72
	CA_EMG1/EMG2	42	2	3.03	0.22	2.59	3.47	0.80	7.55	0.94	0.36	1.20	0.72
	BD_EMG1/EMG2	42	2	2.37	0.23	1.90	2.84	0.19	7.21	1.23	0.36	1.81	0.72
	DB_EMG1/EMG2	42	2	2.54	0.24	2.05	3.03	0.33	8.22	1.45	0.36	3.01	0.72
Oblique	AD_EMG1/EMG2	42	2	2.71	0.24	2.24	3.19	0.42	7.32	0.89	0.36	1.03	0.72
	DA_EMG1/EMG2	42	2	2.46	0.23	1.99	2.92	0.48	6.99	1.15	0.36	1.31	0.72
	BC_EMG1/EMG2	42	2	2.51	0.24	2.03	2.99	0.59	7.90	1.40	0.36	2.59	0.72
	CB_EMG1/EMG2	42	2	2.91	0.28	2.35	3.47	0.64	8.77	1.29	0.36	1.69	0.72
State–Trait	AA_EMG1/EMG2	42	2	2.54	0.21	2.12	2.96	0.80	6.46	1.04	0.36	1.28	0.72
	BB_EMG1/EMG2	42	2	2.24	0.24	1.75	2.74	0.22	7.21	1.35	0.36	1.95	0.72
	CC_EMG1/EMG2	42	2	2.70	0.24	2.22	3.18	0.70	8.95	1.68	0.36	5.42	0.72
	DD_EMG1/EMG2	42	2	2.21	0.23	1.74	2.68	0.32	6.49	1.17	0.36	1.02	0.72

^1^ The letters A, B, C, and D refer to the 4 quadrants of Russell’s Circumplex model, respectively. A corresponds to states of Stress, B corresponds to states of Excitement, C corresponds to states of Relaxation, and D corresponds to states of Boredom. The transitions shown in Table 1 refer to the shifts between these four states. Also shown are the state–trait indices, representative of psychophysiological activation when the subject is in one of the 4 states (AA, BB, CC, DD). AB and BA = transition from Stress to Excitement and vice versa. CD and DC = transition from Boredom to Relaxation and vice versa. AC and CA = transition from Stress to Boredom and vice versa. BD and DB = transition from Excitement to Relaxation and vice versa. AD and DA = transition from Stress to Relaxation and vice versa. BC and CB = transition from Excitement to Boredom and vice versa.

**Table 3 healthcare-12-01690-t003:** Correlation matrix. Facial EMG1/EMG2 ratio’s transitions, CI confidence interval (95% CI are indicated as the inferior and superior values of the interval), *ρ* Spearman’s correlation coefficient, *p*-value.

Self-Report Measure ^1^	EMG1/EMG2 Amplitude (ms^2^) Transition	Correlation Value	95% CI
BDI	AB (from Stress to Engagement)	*ρ* = 0.317; *p* < 0.05	[0.008, 0.564]
	CD (from Boredom to Relax)	*ρ* = 0.314; *p* < 0.05	[0.010, 0.579]
	DB (from Relax to Engagement)	*ρ* = 0.364; *p* < 0.05	[0.061, 0.600]
	DA (from Relax to Stress)	*ρ* = 0.320; *p* < 0.05	[−0.012, 0.583]
	BC (from Engagement to Boredom)	*ρ* = 0.319; *p* < 0.05	[0.016, 0.575]
	CB (from Boredom to Engagement)	*ρ* = 0.383; *p* < 0.05	[0.084, 0.624]
Catastrophizing (CDS)	AB (from Stress to Engagement)	*ρ* = 0.349; *p* < 0.05	[0.032, 0.594]
	BD (from Engagement to Relax)	*ρ* = 0.383; *p* < 0.05	[0.050, 0.656]
	DA (from Relax to Stress)	*ρ* = 0.333; *p* < 0.05	[−0.058, 0.617]
	CB (from Boredom to Engagement)	*ρ* = 0.368; *p* < 0.05	[0.063, 0.610]
All-or-Nothing Thinking (CDS)	BD (from Engagement to Relax)	*ρ* = 0.347; *p* < 0.05	[0.050, 0.608]
Should Statements (CDS)	CD (from Boredom to Relax)	*ρ* = 0.316; *p* < 0.05	[−0.004, 0.538]
	DC (from Relax to Boredom)	*ρ* = 0.332; *p* < 0.05	[0.034, 0.601]
	CA (from Boredom to Stress)	*ρ* = 0.307; *p* < 0.05	[−0.003, 0.580]
	DB (from Relax to Engagement)	*ρ* = 0.356; *p* < 0.05	[0.008, 0.564]
	CB (from Boredom to Engagement)	*ρ* = 0.358; *p* < 0.05	[0.033, 0.600]
Minimization (CDS)	CD (from Boredom to Relax)	*ρ* = 0.364; *p* < 0.05	[0.067, 0.627]
	DB (from Relax to Engagement)	*ρ* = 0.407; *p* < 0.01	[0.132, 0.646]
	DC (from Relax to Boredom)	*ρ* = 0.322; *p* < 0.05	[0.016, 0.574]
	DA (from Relax to Stress)	*ρ* = 0.386; *p* < 0.05	[0.110, 0.614]
	BC (from Engagement to Boredom)	*ρ* = 0.312; *p* < 0.05	[0.052, 0.552]
	CB (from Boredom to Engagement)	*ρ* = 0.379; *p* < 0.05	[0.131, 0.597]
CDS	AB (from Stress to Engagement)	*ρ* = 0.318; *p* < 0.05	[0.030, 0.551]
	DB (from Relax to Engagement)	*ρ* = 0.314; *p* < 0.05	[0.003, 0.572]
	CB (from Boredom to Engagement)	*ρ* = 0.318; *p* < 0.05	[0.049, 0.562]

^1^ Acronyms: BDI stands for Beck Depression Inventory; CDS refers to Cognitive Distortion Scale.

**Table 4 healthcare-12-01690-t004:** Correlation matrix. Facial EMG1/EMG2 ratio during state–trait, CI confidence interval (95% CI are indicated as the inferior and superior values of the interval), *ρ* Spearman’s correlation coefficient, *p*-value.

Self-Report Measure ^1^	EMG1/EMG2 Amplitude (ms^2^) State–Trait	Correlation Value	95% CI
BDI	BB (Engagement States)	*ρ* = 0.312; *p* < 0.05	[−0.003, 0.394]
CDS	DD (Relaxation States)	*ρ* = 0.373; *p* < 0.05	[0.075, 0.636]
Should Statements (CDS)	DD (Relaxation States)	*ρ* = 0.426; *p* < 0.01	[0.109, 0.667]
All-or-Nothing Thinking (CDS)	DD (Relaxation States)	*ρ* = 0.437; *p* < 0.01	[0.173, 0.642]
Minimization (CDS)	BB (Engagement States)	*ρ* = 0.383; *p* < 0.05	[0.100, 0.616]
	DD (Relaxation States)	*ρ* = 0.391; *p* < 0.05	[0.119, 0.604]
Catastrophizing (CDS)	BB (Engagement States)	*ρ* = 0.372; *p* < 0.05	[0.074, 0.621]
	DD (Relaxation States)	*ρ* = 0.352; *p* < 0.05	[0.056, 0.599]

^1^ Acronyms: BDI stands for Beck Depression Inventory; CDS refers to Cognitive Distortion Scale.

## Data Availability

The datasets used and/or analyzed during the current study are available from the corresponding author upon reasonable request.

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
