# Peer review of "Linking Affect Dynamics and Well-Being: A Novel Methodological Approach for Mental Health"

_healthcare, 2024, doi:10.3390/healthcare12171690_

Round 1

Reviewer 1 Report

Comments and Suggestions for Authors

General comment: The present study examined the associations between various psychological characteristics and trends in fEMG activity during affective transitions. I really appreciated the integration of data from self-reports and physiological measures. The topic is very interesting and important. The manuscript is well-written and well-structured, with strong theoretical foundations and promising hints for future research in the field. Thus, I think this paper will be suitable for publication after some minor revisions.

I have just a few comments and suggestions that the Authors can consider to further improve the manuscript.

Comments and suggestions:

- I suggest proving more information about the sample recruitment strategy (online, in person, where, etc).

- I suggest reporting the range of the age of participants and describing the age distribution if relevant (unimodal, bimodal, flat?).

- for the sake of completeness, I suggest considering adding the Cronbach’s Alpha value and the minimum and maximum possible scores for each self-report measure. I imagine that alpha values may not be optimal because of the sample size, despite the scales are strong.

- I guess that the sequence of blocks of stimuli was chosen to have all the possible transitions to study, but I did not find this information explicitly in the manuscript and I suggest adding it to highlight the methodological rigor.

- I am not an expert in EMG or physiological data. I wonder if the EMG procedure requires some data pre-processing (e.g., artifact detection and removal) but I did not find any description about how it was conducted.

- in Table 1 and Table 2, the mean and SD of measures are not very meaningful alone if the reader does not know the scales/measures well. I suggest reporting the minimum and maximum observed scores for each measure (questionnaire, subscale, ratio). Also, the values of skewness and kurtosis for each measure could be useful to have an idea of the distributions.

- It is not clear to me what the authors refer to with the term ‘state-trait’. I suggest that this definition is better underlined in advance, for better understanding by readers unfamiliar with the model and terminology.

- For someone new to the model of Russell, it may be non-trivial to remember the meaning of the quadrants A, B, C, and D. So, I suggest recalling their meaning in Tables 1 and 4 – perhaps in a note below. Also, in Figure 2 and 3, I suggest recalling the meaning of A, B, C, D, for readers.

- I was wondering why it was chosen not to report the non-significant correlations in Tables 3 and 4. Correlations is an effect size, and it is interesting to also observe non-statistically significant values in an exploratory study such as this one. The authors may consider adding them, perhaps in the supplementary materials if they think it is of minor relevance or it makes the main text too long.

- In the limitations section, I suggest mentioning two related points. One is that the effect sizes found in this study are smaller (r around .3) than those used in the power analysis (r = .52), which may imply some limitations. The other is about suggesting caution in the interpretation of results with large 95%CI.

- About future research, I suggest mentioning to conduct studies with specific characteristics that will address the aforementioned limitations.

Minor comments:

- line 62 – the MDD acronym appears but was never defined.

- I would add a note to Table 4 to make explicit acronyms such as CDS. This applies other tables, if needed.

- line 282, there seems to be a typo when you refer to “12 blocks”. If I understood correctly there are 13 blocks and 12 transitions.

- line 288 – there is a typo in Russel(l)

- line 315 - p should be lowercase

- line 338 – there is a repetition

Author Response

Dear Reviewer,

First and foremost, I would like to express our gratitude for the time and effort invested in reviewing my manuscript. Your insightful comments and suggestions have been invaluable in enhancing the quality and clarity of this work. We appreciate the constructive feedback provided, which has significantly contributed to the development of a more robust and well-rounded article. We have carefully considered and addressed each point raised in the revision process.

Thank you once again for your valuable input and guidance.

We highlighted the revisions in green in both the attached reviewer’s answer file and the main manuscript.

General comment: The present study examined the associations between various psychological characteristics and trends in fEMG activity during affective transitions. I really appreciated the integration of data from self-reports and physiological measures. The topic is very interesting and important. The manuscript is well-written and well-structured, with strong theoretical foundations and promising hints for future research in the field. Thus, I think this paper will be suitable for publication after some minor revisions.

I have just a few comments and suggestions that the Authors can consider to further improve the manuscript.

Comments and suggestions:

Comment 1: I suggest proving more information about the sample recruitment strategy (online, in person, where, etc).

Response 1: Thank you for the valuable suggestion!  We have specified more about how subjects have been recruited.

Lines 153-157 (Page 4): “The invitation to participate in the study was conducted using both probabilistic cluster sampling and avalanche sampling methods. The trial organizers reached out to the volunteer participants who had consented to participate in the study via email to coordinate the specific time and place of the experiment. The experimental activity took place in the HST laboratory of the Psychology Department (University of Turin).

Comment 2: I suggest reporting the range of the age of participants and describing the age distribution if relevant (unimodal, bimodal, flat?).

Response 2: Thank you very much for the valuable suggestion. We have included information about the range and age distribution of males and females.

Lines 158-161 (Page 4): “The average age of the males was 26 with a standard deviation of 3.59 (ranging from 21 to 36), while the average age of the women was 24.77 with a standard deviation of 4.99 (ranging from 18 to 43). Both gender groups have a unimodal age distribution, with a single noticeable peak occurring in the mid-20s.”

Comment 3: for the sake of completeness, I suggest considering adding the Cronbach’s Alpha value and the minimum and maximum possible scores for each self-report measure. I imagine that alpha values may not be optimal because of the sample size, despite the scales are strong.

Response 3: thank you for the suggestion! We added Cronbach’s Alpha in Table 1 (line 287). The section presenting the two questionnaires now includes the minimum and maximum possible scores. Respectively, BDI’s minimum and maximum possible scores have been inserted in lines 187-188 (Page 4): “The minimum obtainable score equals 0, while the maximum corresponds to 63”; CDS’s subscales and total scale minimum and maximum have been added in lines 190-192 (Page 4): “Each category consists of 2 items, resulting in a minimum observable score of 2 and a maximum score of 14 for each category. Regarding the total scale (CDS), scores between 20 and 140 can be observed.”

The aforementioned information was not included in Table 1 to allow for the incorporation of supplementary indexes, as recommended in Comment 6.

Comment 4: I guess that the sequence of blocks of stimuli was chosen to have all the possible transitions to study, but I did not find this information explicitly in the manuscript and I suggest adding it to highlight the methodological rigor.

Response 4: Thank you for pointing out this fundamental characteristic of the methodology that we employed! Considering and inducing in participants all the possible transitions that can occur between affective states allowed us to explore their physiological responses in more detail. We have inserted a specification in lines 130 (Page 3) “all the” and 131-133 (Page 3): “Thus, referring to Russell's model, the four affective states provided a basis for identifying 12 potential transitions that may occur between them.”

Comment 5: I am not an expert in EMG or physiological data. I wonder if the EMG procedure requires some data pre-processing (e.g., artifact detection and removal) but I did not find any description about how it was conducted.

Response 5: Thank you for asking this specific! We provided a more detailed explanation of the processing phase in lines 267-270: “A 50 Hz notch filter was applied to remove power line interference. Subsequently, amplitude filtering was performed within the 20-500 Hz range. Additionally, a visual inspection was conducted to identify and remove any remaining artifacts.”

Comment 6:  in Table 1 and Table 2, the mean and SD of measures are not very meaningful alone if the reader does not know the scales/measures well. I suggest reporting the minimum and maximum observed scores for each measure (questionnaire, subscale, ratio). Also, the values of skewness and kurtosis for each measure could be useful to have an idea of the distributions.

Response 6: Thank you for bringing this issue to our attention. We have added in Tables 1 and 2 the requested parameters: minimum and maximum observed scores, skewness and Kurtosis values. You will find them in line 304 (Page 7) for Table 1 and in line 306 (Page 8) for Table 2.

Adding the latter values, we observed that although the distribution of the scores obtained on the scales is approximately normal, the same cannot be said of the distributions of the values of the physiological indices. In fact, the electrical activity produced by the contraction of facial muscles can vary greatly from one subject to another. Therefore, thanks to your suggestion, we have updated the correlation matrix using Spearman's correlation coefficient. It is possible to observe them in Table 3 (line 325) and 4 (line 353). The use of Spearman’s correlation revealed additional correlations between the questionnaires and psychophysiological data, consistent with our previous findings. As a result, these observations have been included in the Results and Discussion sections. Below are the lines in which to find the relevant considerations.

Results section:

  • Line 330 (Page 10): “The correlation between the BDI and AB was r = 0.349 (p < 0.05)”.
  • Lines 350-351 (Page 11): “CDS total score showed significant correlations with three affective transitions: with AB (r = 0.373, p < 0.05), with DB (r  = 0.444, p < 0.01) and with CB (r  = 0.318, p < 0.05).”
  • Line 360 (Page 12): “BDI scores displayed a correlation with DD state-trait (r = 0.312, p < 0.05).”
  • Lines 362-363 (Page 11): “Should Statements (CDS) show a more consistent correlation with DD state-trait, with r = 0.426 and p < 0.01”
  • Lines 366-368 (Page 12): “Catastrophizing (CDS) scores emerged correlated with two state-trait conditions: the first one is BB (r = 0.372, p < 0.05) and the second one is DD (r = 0.352, p < 0.05); both correlations reflected a moderate positive relationship.”

Discussion section:

  • Lines 379-381 (Page 12): “CD (transition from poorly activating negative emotional states to poorly activating positive emotional states, e.g. from boredom to relaxation)”
  • Lines 382-384 (Page 12): “DA (transition from poorly activating positive emotional state to highly activating negative emotional state, e.g. from relaxation to stress)
  • Lines 3958-408 (Page 12): “Notable correlations have been found between the overall score on the CDS scale and psychophysiological activations during three transitions: AB (transition from stress to engagement), DB (transition from relax to engagement) and CB (transition from boredom to engagement). Once more, the presence of extremely activating emotional states is noticeable (B). Prior studies in the literature [47], [48] have shown the association between individuals with depression who engage in cognitive distortions and the manifestation of alterations in facial muscle activation. These modifications are often associated with the automatic processing of emotional inputs, whereby individuals with depression tend to process negative facial expressions more quickly and readily, whereas delighted expressions take more cognitive effort and are less likely to be performed”
  • Lines 422-424 (Page 13): “BD (from a highly activating positive emotional state to a slightly activating positive emotional state, e.g. from engagement to relax”
  • Line 430 (Page 13): “… but also between positive states (BD)”
  • Lines 451-452 (Page 13): “CD (from a slightly activating negative emotional state to a slightly activating positive emotional state, e.g., from boredom to relax)”
  • Lines 457-458 (Page 13): “CB (from a slightly activating negative emotional state to a highly activating positive emotional state, e.g., from boredom to engagement).”
  • Lines 476-477 (Page 14): “DC (from low activating positive emotional state to low activating negative emotional state, e.g., from relax to boredom)”
  • Lines 479-480 (Page 14): “BC from highly activating positive emotional state to poorly activating negative emotional state, e.g., from arousal to boredom)”
  • Lines 489-491 (Page 14): “The overall score on the BDI scale was shown to be correlated with the EMG1/EMG2 ratio in the BB condition, which corresponds to a positive emotional state with a high level of activation.”
  • Lines 494-498 (Page 14): “In our study, “All or Nothing Thinking” had a positive correlation with EMG1/EMG2 ratio during BB states. What emerged is consistent with previous literature emphasizing the characteristics of rigidity, disappointment and loss perceived by individuals who implement this cognitive distortion [52].”

Comment 7: It is not clear to me what the authors refer to with the term ‘State-trait’. I suggest that this definition is better underlined in advance, for better understanding by readers unfamiliar with the model and terminology.

Response 7: Thank you very much for bringing this to our attention! it is indeed of paramount importance that the reader can follow as clearly as possible what is reported in our study. We have therefore taken care to specify the notion of state-trait. You will find it in the lines 133-137 (Page 3): “Furthermore, alongside the transition indexes state-trait indexes were also selected for extrapolation. These measures represent the subject's psychophysiological activity when experiencing a particular affective state. By observing these indexes, we may get information on the consistency of the emotional states that the individual is experiencing.” And in line 245 (Page 6): “corresponding to the state-trait indexes”.

Comment 8: For someone new to the model of Russell, it may be non-trivial to remember the meaning of the quadrants A, B, C, and D. So, I suggest recalling their meaning in Tables 1 and 4 – perhaps in a note below. Also, in Figure 2 and 3, I suggest recalling the meaning of A, B, C, D, for readers.

Response 8: Thank you so much for providing this vital observation! Again, it is crucial for us that the reader can access information as quickly and easily as possible about the association between the letters in the four quadrants and the affective state represented. We have taken care to add such specifics in several places in the paper.

Note to Table 2 (line 306, Page 8): “The letters A, B, C, D refer to the 4 quadrants of Russell's Circumplex model, respectively. A corresponds to states of Stress, B corresponds to states of Excitement, C corresponds to states of Relaxation, and D corresponds to states of Boredom. The transitions shown in Table 1 refer to the shifts between these four states. Also shown are the state-trait indices, representative of psychophysiological activation when the subject is in one of the 4 states (AA, BB, CC, DD).

AB and BA = transition from Stress to Excitement and vice versa. CD and DC = transition from Boredom to Relaxation and vice versa. AC and CA = transition from Stress to Boredom and vice versa. BD and DB = transition from Excitement to Relaxation and vice versa. AD and DA = transition from Stress to Relaxation and vice versa. BC and CB = transition from Excitement to Boredom and vice versa.”

Lines 251-252 (Figure 2, Page 6): “Stress is represented by A blocks, Excitement is represented by B blocks, Boredom is represented by C blocks, and Relaxation is represented by D blocks”.

Lines 317-319 (Figure 3, Page 9): “A blocks represent Stress, B blocks represent Excitement, C blocks represent Boredom and D blocks represent Relaxation.”

Line 351 (Table 4, Page 11).

Comment 9: I was wondering why it was chosen not to report the non-significant correlations in Tables 3 and 4. Correlations is an effect size, and it is interesting to also observe non-statistically significant values in an exploratory study such as this one. The authors may consider adding them, perhaps in the supplementary materials if they think it is of minor relevance or it makes the main text too long.

Response 9: Thank you for this suggestion! We have also included the correlations that are not statistically significant in the "Supplementary materials" section, so as not to make the text heavy, as you suggested. References can be found in lines 318-319: “(non-significant correlations can be observed in Table S3 and Table S4 in Supplementary Materials).”

Comment 10: In the limitations section, I suggest mentioning two related points. One is that the effect sizes found in this study are smaller (r around .3) than those used in the power analysis (r = .52), which may imply some limitations. The other is about suggesting caution in the interpretation of results with large 95%CI.

Response 10: Thank you for bringing this issue to our attention; what you mentioned is critical for directing future studies keeping in mind the sample size and effect size. We have included suggested limitations in a separated section “Limitations” (line 559).

Lines 563-565 (page 15): “It is also necessary to mention that the effect sizes observed in the present study are smaller (in most cases, ρ ≈ 0.3) than those used to establish the sample size in the Power Analysis (r = 0.52) and CI are large, due to intraindividual variability in psychophysiological parameters. Future studies should consider these limitations and address them to enhance statistical power.”

Comment 11: About future research, I suggest mentioning to conduct studies with specific characteristics that will address the aforementioned limitations.

Response 11: Thank you very much for the valuable suggestion! Immediately after mentioning the suggested limitations, we included a section concerning future studies.

Lines 566-567 (Page 15): “Future studies should consider these limitations and address them to enhance statistical power”.

Minor comments:

- line 62 – the MDD acronym appears but was never defined.

Thank you or bringing it to our attention! We inserted the extended definition: line 62: Major Depressive Disorder (MDD

- I would add a note to Table 4 to make explicit acronyms such as CDS. This applies other tables, if needed.

Thank you for the suggestion! It is important that the presence of acronyms does not hinder the reader's understanding. We inserted a note for Table 1, 3 and 4. “Acronyms: BDI stands for Beck Depression Inventory; CDS refers to Cognitive Distortion Scale”.

- line 282, there seems to be a typo when you refer to “12 blocks”. If I understood correctly there are 13 blocks and 12 transitions.

Thank you for bringing it to our attention! We corrected it in line 311 (page 9).

- line 288 – there is a typo in Russel(l)

Thank you or bringing it to our attention! We corrected it!

- line 315 - p should be lowercase

Thank you or bringing it to our attention! We corrected it!

- line 338 – there is a repetition

Thank you or bringing it to our attention! We corrected it!

Reviewer 2 Report

Comments and Suggestions for Authors

Thank you for the opportunity to review this intriguing paper, which explores the relationship between affect dynamics and well-being using facial electromyography (fEMG) methodology.

Overall, I find the manuscript to be well-written, logically organized, and comprehensive in its presentation of the key content.

  • The abstract is concise, informative, and appropriately detailed.
  • The introduction effectively presents the theoretical background and foundational concepts. However, I recommend the authors consider revising the order of the final three paragraphs. It seems that the concluding paragraph might be more appropriately placed earlier in the text (so that the introduction ends with the purpose of the study).
  • The methods and materials section is clear and comprehensible. One aspect that could benefit from clarification is whether all participants were exposed to the stimuli in the same sequence or if they were randomly assigned to groups with varying orders. Could this procedural detail potentially influence the results? I suggest the authors provide further explanation on this point.
  • The results are well presented. However, I would like to understand the criteria the authors used in deciding to employ the Pearson correlation coefficient. Did they assess the normality of the variable distributions and other relevant assumptions that justify the use of this statistical measure? Clarification on this matter would be beneficial.
  • The authors mention that only significant correlations are reported in the results. It might be valuable to include the full correlation matrix as an appendix, allowing readers to examine the magnitude of the other correlation coefficients.
  • In the results section (lines 303–341), the authors frequently use terms like "indicates" and "suggest" to interpret the findings. It might be more appropriate to relocate these interpretations to the discussion section, where the findings can be fully contextualized.
  • The discussion and conclusions are clear and well-reasoned.

Overall, this paper makes a significant contribution to the field and is well-structured and informative.

Author Response

Dear Reviewer,

First and foremost, I would like to express our gratitude for the time and effort invested in reviewing my manuscript. Your insightful comments and suggestions have been invaluable in enhancing the quality and clarity of this work. We appreciate the constructive feedback provided, which has significantly contributed to the development of a more robust and well-rounded article. We have carefully considered and addressed each point raised in the revision process.

Thank you once again for your valuable input and guidance.

We highlighted the revisions in green in both the attached reviewer’s answer file and the main manuscript.

Thank you for the opportunity to review this intriguing paper, which explores the relationship between affect dynamics and well-being using facial electromyography (fEMG) methodology.

Reviewer’s Comments

Overall, I find the manuscript to be well-written, logically organized, and comprehensive in its presentation of the key content.

Comment 1: The abstract is concise, informative, and appropriately detailed.

Response 1: Thank you so much!

Comment 2: The introduction effectively presents the theoretical background and foundational concepts. However, I recommend the authors consider revising the order of the final three paragraphs. It seems that the concluding paragraph might be more appropriately placed earlier in the text (so that the introduction ends with the purpose of the study).

Response 2: We much appreciate your feedback about the arrangement of the concluding paragraphs in the Introduction section. It is crucial to provide the reader with a thorough introduction to the topic while maintaining the study's emphasis. As per your advice, we have rearranged the organization of the last paragraphs. Firstly, facial electromyography is introduced as a research tool for studying depressed symptoms (from line 89, page 2). The significance of studying subclinical symptoms is then highlighted (from line 108, page 3). Finally, the suggested approach for the current study is described (from line 117 and from line 138, page 3).

Comment 3: The methods and materials section is clear and comprehensible. One aspect that could benefit from clarification is whether all participants were exposed to the stimuli in the same sequence or if they were randomly assigned to groups with varying orders. Could this procedural detail potentially influence the results? I suggest the authors provide further explanation on this point.

Response 3: Thank you for the question raised! Each participant was given a unique, random sequence of images to avoid time or ceiling effects in the emotional response. This design ensured a full exploration of emotional transitions, with the sequence of image presentation randomized to avoid order effects.

Under your suggestion, we included this information in section 2.3.2 Psychophysiological evaluation in lines 237-240 (Page 5 and 6): “To prevent a temporal or ceiling impact in their emotional reaction, each participant saw a different series of randomly selected pictures. This method avoided order effects by presenting the images in a randomized order to allow for a thorough investigation of emotional transitions.”

Comment 4: The results are well presented. However, I would like to understand the criteria the authors used in deciding to employ the Pearson correlation coefficient. Did they assess the normality of the variable distributions and other relevant assumptions that justify the use of this statistical measure? Clarification on this matter would be beneficial.

Response 4: Thank you for bringing this issue to our attention! We performed a recheck about the distribution of data. While the data from the questionnaires follow a normal distribution, the data inherent in the fEMG signals do not. In fact, the electrical activity produced by the contraction of facial muscles can vary greatly from one subject to another. Therefore, thanks to your suggestion, we have updated the correlation matrix using Spearman's correlation coefficient. In order to provide as much clear and explicit information as possible, we have also included skewness and Kurtosis values in Tables 1 and 2 (lines 301, 303, Page 7, 8). Consequently, we updated the correlation coefficients and their confidence intervals in Table 3 and 4 (lines 325 and 353, page 10 and 11). The use of Spearman’s correlation revealed additional correlations between the questionnaires and psychophysiological data, consistent with our previous findings. As a result, these observations have been included in the Results and Discussion sections. Below are the lines in which to find the relevant considerations.

Results section:

  • Line 330 (Page 10): “The correlation between the BDI and AB was r = 0.349 (p < 0.05)”.
  • Lines 350-351 (Page 11): “CDS total score showed significant correlations with three affective transitions: with AB (r = 0.373, p < 0.05), with DB (r  = 0.444, p < 0.01) and with CB (r  = 0.318, p < 0.05).”
  • Line 360 (Page 12): “BDI scores displayed a correlation with DD state-trait (r = 0.312, p < 0.05).”
  • Lines 362-363 (Page 11): “Should Statements (CDS) show a more consistent correlation with DD state-trait, with r = 0.426 and p < 0.01”
  • Lines 366-368 (Page 12): “Catastrophizing (CDS) scores emerged correlated with two state-trait conditions: the first one is BB (r = 0.372, p < 0.05) and the second one is DD (r = 0.352, p < 0.05); both correlations reflected a moderate positive relationship.”

Discussion section:

  • Lines 379-381 (Page 12): “CD (transition from poorly activating negative emotional states to poorly activating positive emotional states, e.g. from boredom to relaxation)”
  • Lines 382-384 (Page 12): “DA (transition from poorly activating positive emotional state to highly activating negative emotional state, e.g. from relaxation to stress)
  • Lines 3958-408 (Page 12): “Notable correlations have been found between the overall score on the CDS scale and psychophysiological activations during three transitions: AB (transition from stress to engagement), DB (transition from relax to engagement) and CB (transition from boredom to engagement). Once more, the presence of extremely activating emotional states is noticeable (B). Prior studies in the literature [47], [48] have shown the association between individuals with depression who engage in cognitive distortions and the manifestation of alterations in facial muscle activation. These modifications are often associated with the automatic processing of emotional inputs, whereby individuals with depression tend to process negative facial expressions more quickly and readily, whereas delighted expressions take more cognitive effort and are less likely to be performed”
  • Lines 422-424 (Page 13): “BD (from a highly activating positive emotional state to a slightly activating positive emotional state, e.g. from engagement to relax”
  • Line 430 (Page 13): “… but also between positive states (BD)”
  • Lines 451-452 (Page 13): “CD (from a slightly activating negative emotional state to a slightly activating positive emotional state, e.g., from boredom to relax)”
  • Lines 457-458 (Page 13): “CB (from a slightly activating negative emotional state to a highly activating positive emotional state, e.g., from boredom to engagement).”
  • Lines 476-477 (Page 14): “DC (from low activating positive emotional state to low activating negative emotional state, e.g., from relax to boredom)”
  • Lines 479-480 (Page 14): “BC from highly activating positive emotional state to poorly activating negative emotional state, e.g., from arousal to boredom)”
  • Lines 489-491 (Page 14): “The overall score on the BDI scale was shown to be correlated with the EMG1/EMG2 ratio in the BB condition, which corresponds to a positive emotional state with a high level of activation.”
  • Lines 494-498 (Page 14): “In our study, “All or Nothing Thinking” had a positive correlation with EMG1/EMG2 ratio during BB states. What emerged is consistent with previous literature emphasizing the characteristics of rigidity, disappointment and loss perceived by individuals who implement this cognitive distortion [52].”

Comment 5: The authors mention that only significant correlations are reported in the results. It might be valuable to include the full correlation matrix as an appendix, allowing readers to examine the magnitude of the other correlation coefficients.

Response 5: Thank you for this suggestion! We have also included the correlations that are not statistically significant in the "Supplementary materials" section, so as not to make the text heavy, as you suggested.

Comment 6: In the results section (lines 303–341), the authors frequently use terms like "indicates" and "suggest" to interpret the findings. It might be more appropriate to relocate these interpretations to the discussion section, where the findings can be fully contextualized.

Response 6: Thank you for the valuable suggestion! We have removed from the Results section all the parts that might resemble an interpretation of the results that emerged. As you rightly noted, it is more appropriate to keep observations and considerations only in the Discussion section.

Comment 7: The discussion and conclusions are clear and well-reasoned.

Response 7: Thank you again for your vital contribution!

Overall, this paper makes a significant contribution to the field and is well-structured and informative.

Reviewer 3 Report

Comments and Suggestions for Authors

The authors propose an innovative methodology to detect the early symptoms of mental illnesses such as major depression. The topic is of undoubted interest and potentially relevant for early diagnosis. The authors hypothesise that, depending on the propensity or otherwise to manifest depressive symptoms, subjects show different results in the electrical activity of certain facial muscles known to be connected with the manifestation of emotions. 

The methodological framework of the experiment is particularly accurate and constructed by referring to Russell's Circumflex model of emotions.

However, some aspects of data processing merit further clarification. To test their hypothesis, the authors rely on a correlation matrix that predicts the normal distribution of the data subjected to analysis. But no measures (statistical or graphical) are reported to prove the normality of the data. The presence of a few large standard deviations (Table 1 BDI score and total CDS score) does not seem to testify with certainty to the normality of the distribution of the scores. Furthermore, the convenience sample used initially has a numerosity of 44 (Table 1) then (Table 2) decreases to 42 in line with the minimum number of the estimated sample numerosity assessed a priori. It would be useful to know based on which criteria two subjects were excluded from the group (they could be outliers whose exclusion might have altered the distribution of scores) but, in any case, reporting two different numerosities in two adjoining tables is confusing without adequate explanation.  Another curiosity is represented by the use of an ethics committee of a body other than the one to which it belongs: all medium-large Italian universities, and Turin is one, have an ethics committee to which research projects are submitted but the authors, who belong to the Department of Psychology of that university, they turned to the Ethics Committee of a well-known institute operating in the same region.

As regards the results, the correlations, on which the authors base the connection between electrical activity of the facial muscles and propensity to depression, are all of moderate magnitude (they explain a maximum of 15% of the variability) but are statistically significant. The graphic representation of figure 3 could be improved for example by placing the labels with the values ​​relating to the states with a positive valence all below the line and those with the values ​​of the states with a negative valence above the line: the current placement does not seem to make sense. Lastly, there appears to be a lack of explanation from the authors regarding the possible application of the results to the diagnostic process. To what extent can fEMG be useful in the diagnostic phase and to what extent can it be preferable, in terms of precision and earliness, but also for ease of use, to the most widespread diagnostic tools such as questionnaires?

Comments on the Quality of English Language

The construction of the text reflects the construction of the authors' language. We recommend having the text reviewed by a native speaker

Author Response

Dear Reviewer,

First and foremost, I would like to express our gratitude for the time and effort invested in reviewing my manuscript. Your insightful comments and suggestions have been invaluable in enhancing the quality and clarity of this work. We appreciate the constructive feedback provided, which has significantly contributed to the development of a more robust and well-rounded article. We have carefully considered and addressed each point raised in the revision process.

Thank you once again for your valuable input and guidance.

We highlighted the revisions in green in both the attached reviewer’s answer file and the main manuscript.

Reviewer’s comments

The authors propose an innovative methodology to detect the early symptoms of mental illnesses such as major depression. The topic is of undoubted interest and potentially relevant for early diagnosis. The authors hypothesise that, depending on the propensity or otherwise to manifest depressive symptoms, subjects show different results in the electrical activity of certain facial muscles known to be connected with the manifestation of emotions. 

The methodological framework of the experiment is particularly accurate and constructed by referring to Russell's Circumflex model of emotions.

However, some aspects of data processing merit further clarification.

Comment 1: To test their hypothesis, the authors rely on a correlation matrix that predicts the normal distribution of the data subjected to analysis. But no measures (statistical or graphical) are reported to prove the normality of the data. The presence of a few large standard deviations (Table 1 BDI score and total CDS score) does not seem to testify with certainty to the normality of the distribution of the scores.

Response 1: Thank you for bringing this issue to our attention! We performed a recheck about the distribution of data. While the data from the questionnaires follow a normal distribution, the data inherent in the fEMG signals do not. In fact, the electrical activity produced by the contraction of facial muscles can vary greatly from one subject to another. Therefore, thanks to your suggestion, we have updated the correlation matrix using Spearman's correlation coefficient. In order to provide as much clear and explicit information as possible, we have also included skewness and Kurtosis values in Tables 1 and 2 (lines 304, 306). Consequently, we updated the correlation coefficients and their confidence intervals in Table 3 and 4 (lines 325 and 353). The use of Spearman’s correlation revealed additional correlations between the questionnaires and psychophysiological data, consistent with our previous findings. As a result, these observations have been included in the Results and Discussion sections. Below are the lines in which to find the relevant considerations.

Results section:

  • Line 330 (Page 10): “The correlation between the BDI and AB was r = 0.349 (p < 0.05)”.
  • Lines 350-351 (Page 11): “CDS total score showed significant correlations with three affective transitions: with AB (r = 0.373, p < 0.05), with DB (r  = 0.444, p < 0.01) and with CB (r  = 0.318, p < 0.05).”
  • Line 360 (Page 12): “BDI scores displayed a correlation with DD state-trait (r = 0.312, p < 0.05).”
  • Lines 362-363 (Page 11): “Should Statements (CDS) show a more consistent correlation with DD state-trait, with r = 0.426 and p < 0.01”
  • Lines 366-368 (Page 12): “Catastrophizing (CDS) scores emerged correlated with two state-trait conditions: the first one is BB (r = 0.372, p < 0.05) and the second one is DD (r = 0.352, p < 0.05); both correlations reflected a moderate positive relationship.”

Discussion section:

  • Lines 379-381 (Page 12): “CD (transition from poorly activating negative emotional states to poorly activating positive emotional states, e.g. from boredom to relaxation)”
  • Lines 382-384 (Page 12): “DA (transition from poorly activating positive emotional state to highly activating negative emotional state, e.g. from relaxation to stress)
  • Lines 3958-408 (Page 12): “Notable correlations have been found between the overall score on the CDS scale and psychophysiological activations during three transitions: AB (transition from stress to engagement), DB (transition from relax to engagement) and CB (transition from boredom to engagement). Once more, the presence of extremely activating emotional states is noticeable (B). Prior studies in the literature [47], [48] have shown the association between individuals with depression who engage in cognitive distortions and the manifestation of alterations in facial muscle activation. These modifications are often associated with the automatic processing of emotional inputs, whereby individuals with depression tend to process negative facial expressions more quickly and readily, whereas delighted expressions take more cognitive effort and are less likely to be performed”
  • Lines 422-424 (Page 13): “BD (from a highly activating positive emotional state to a slightly activating positive emotional state, e.g. from engagement to relax”
  • Line 430 (Page 13): “… but also between positive states (BD)”
  • Lines 451-452 (Page 13): “CD (from a slightly activating negative emotional state to a slightly activating positive emotional state, e.g., from boredom to relax)”
  • Lines 457-458 (Page 13): “CB (from a slightly activating negative emotional state to a highly activating positive emotional state, e.g., from boredom to engagement).”
  • Lines 476-477 (Page 14): “DC (from low activating positive emotional state to low activating negative emotional state, e.g., from relax to boredom)”
  • Lines 479-480 (Page 14): “BC from highly activating positive emotional state to poorly activating negative emotional state, e.g., from arousal to boredom)”
  • Lines 489-491 (Page 14): “The overall score on the BDI scale was shown to be correlated with the EMG1/EMG2 ratio in the BB condition, which corresponds to a positive emotional state with a high level of activation.”
  • Lines 494-498 (Page 14): “In our study, “All or Nothing Thinking” had a positive correlation with EMG1/EMG2 ratio during BB states. What emerged is consistent with previous literature emphasizing the characteristics of rigidity, disappointment and loss perceived by individuals who implement this cognitive distortion [52].”

Comment 2: Furthermore, the convenience sample used initially has a numerosity of 44 (Table 1) then (Table 2) decreases to 42 in line with the minimum number of the estimated sample numerosity assessed a priori. It would be useful to know based on which criteria two subjects were excluded from the group (they could be outliers whose exclusion might have altered the distribution of scores) but, in any case, reporting two different numerosities in two adjoining tables is confusing without adequate explanation.  

Response 2: Thank you for your observation, you are absolutely right! it is critical not to create unexplained inconsistencies within the paper. Due to misregistration of the psychophysiological fEMG signal, which emerged during the processing phase, 2 subjects reported a very dirty signal, characterized by electrical noise. They were therefore considered as missing data in table 2. Since they had completed the self-report questionnaires, however, their responses were considered in the calculation of the indexes of the various scales. In order to make this clear to the reader, we have included this information in the paper.

Lines 272-275: “Two individuals reported a highly filthy signal with electrical noise because the psychophysiological fEMG signal, which emerged during the processing phase, was incorrectly recorded. As a result, they were regarded as missing data and included in Table 2 as "missing".”

Comment 3: Another curiosity is represented by the use of an ethics committee of a body other than the one to which it belongs: all medium-large Italian universities, and Turin is one, have an ethics committee to which research projects are submitted but the authors, who belong to the Department of Psychology of that university, they turned to the Ethics Committee of a well-known institute operating in the same region.

Response 3: I appreciate you asking this question! The University of Turin has an agreement with Istituto Auxologico Italiano Piancavallo, so when studies involve patients, the trial is conducted in Auxologico; when they involve healthy subjects, it is carried out in Turin at the laboratories of the Department of Psychology.

Comment 4: As regards the results, the correlations, on which the authors base the connection between electrical activity of the facial muscles and propensity to depression, are all of moderate magnitude (they explain a maximum of 15% of the variability) but are statistically significant. The graphic representation of figure 3 could be improved for example by placing the labels with the values ​​relating to the states with a positive valence all below the line and those with the values ​​of the states with a negative valence above the line: the current placement does not seem to make sense.

Response 4: Thank you for pointing out the inconsistencies in Figure 3. The figures should help the reader in understanding the study, so, under your suggestion, we have updated Figure 3 (line 315, page 9) by inserting the labels of the values of the negative states above the line and those of the values of the positive states below the line.

Comment 5: Lastly, there appears to be a lack of explanation from the authors regarding the possible application of the results to the diagnostic process. To what extent can fEMG be useful in the diagnostic phase and to what extent can it be preferable, in terms of precision and earliness, but also for ease of use, to the most widespread diagnostic tools such as questionnaires?

Response 5: Thank you for raising this issue! Our suggested approach seeks to reduce certain biases associated with the questionnaires' design by combining psychophysiological and self-report measurements. Questionnaires may in fact be subject to reporting bias and strictly depend on the subject's awareness of their emotional states and their willingness to express them. The exclusive use of self-report questionnaires may consequently result in an underestimating of the individual's state, particularly in situations of minor symptoms and sub-clinical conditions in which just signals and not a full-blown disease may be present. Furthermore, a person may not identify certain signs or symptoms in the early or prodromal phases of the illness. These issues could be addressed by using a reliable instrument like fEMG, which alterations have been linked to both clinical and sub-clinical depression. Using these two approaches together might be very beneficial, first in the realm of research and subsequently in the clinic. Still, further advancements are required. We added these considerations, under your suggestion, in lines 536-544: “We proposed a method that combined self-report questionnaires with psychophysiological measures, such as fEMG. While self-report methods may be prone to reporting bias and dependent on the subject's awareness and willingness to express their emotional states, the inclusion of fEMG within the diagnostic process could provide a more objective assessment [64]. This approach could be particularly valuable in identifying minor symptoms and sub-clinical conditions that might not be fully captured through questionnaires alone. The employment of fEMG, which has been linked to both clinical and sub-clinical depression, could enhance the detection of early signs or symptoms that might otherwise go unnoticed.”
